# Simplification of 2D shear wave elastography by enlarged SWE box and multiple regions of interest in one acquisition

**Moritz Weiher** [ORCID]*, **Felix Gunnar Richtering, Yvonne Dörffel, Hans-Peter Müller**

Medical Outpatient Department, Charité Universitätsmedizin Berlin, Berlin, Germany

\* moritz.weiher@charite.de

## Abstract

Two-dimensional (2D) Shear Wave Elastography (SWE) is an easy to perform technique to evaluate the liver stiffness. To simplify the procedure and reduce the acquisition time we enlarged the size of the SWE-box and set ten regions of interest (ROI) in one acquisition. We compare the accuracy of this method to ten separate acquisitions in a small box each with a single ROI measurement. Sixty-nine volunteers with diffuse chronic liver disease were studied with 2D-SWE using a Canon Aplio i800 ultrasound system. The shear-wave-speed was measured in the right lobe in ten separate acquisitions and compared to one acquisition with increased size of the SWE-box and ten different ROI measurements. A Bland-Altmann plot was drawn and the interclass correlation coefficient (ICC) was calculated to compare both methods. Finally, 2D-SWE was successfully performed thru both methods in sixty-six participants. Between both methods the ICC is 0.82. The results of this study show a good reliability between ten separate measures and one grouped measure with ten ROI if the mean is below 1.6m/s (7.7kPa). For higher degrees of fibrosis (≥F2) further investigations are needed.

**Data Availability Statement:** All relevant data are within the paper and its Supporting Information files.

## Introduction

The management of patients with chronic liver disease requires easier and harmless approaches to monitor disease progressions [1]. In the last two decades many ultrasound-based methods and systems are available to follow the fibrotic changes in the liver [2–4]. The best validated method is the transient elastography [5, 6], which is easy to perform but is neither guided by B-mode picture nor enables the user to choose the area of measurement because of fixed settings. Some of these problems can be avoided by using two-dimensional (2D) shear wave elastography (SWE), which has become more available and popular in the last years [7, 8].

An advantage of 2D-SWE is the possibility to see the elasticity in an area of tissue as a coloured map inside the SWE-box, which is also named analysis-box or field-of-view and can be adjusted in size and location by the operator. It is possible to measure the elasticity in different, operator-chosen areas, inside the SWE-box by using different regions of interest (ROI). Chung

**Funding:** The authors received no specific funding for this work.

**Competing interests:** The authors have declared that no competing interests exist.

et al. recently compared five separate acquired elasticity maps with single ROI measurements with five so-called grouped measurements in two separate acquired elasticity maps with two to three ROIs in one small SWE-box in the liver [9].

The operator can increase the size of the SWE-box. This allows seeing the elasticity of a greater liver area, shortens the examination time by setting more ROI in the SWE-box and makes the examination easier. On the other hand, it causes an increase in artefacts.

In this study we compare ten separate acquisitions each with one ROI to one acquisition with an increased size of the SWE-box containing ten ROI.

## Materials and methods

### Study population

Our institutional Ethics Committee approved the study (case number: EA1/206/19) and all participants gave written informed consent. This prospective study includes 69 patients who underwent an ultrasound examination because of a known diffuse chronic liver disease in our outpatient department. The age, sex, weight, height and cause of liver disease were recorded for every participant. The participants were asked not to eat three hours before the examination [10].

### Equipment and methods

2D SWE studies were performed using the Aplio i800 ultrasound system of Canon (Canon Medical Systems, Tokyo, Japan) with a convex broadband probe (i8CX1). The pre-set "Fibrosis i8CX1" was selected before the SWE mode was started. The physical and technical specifications of SWE are written in detail elsewhere [2, 11]. The examinations were performed on the right lobe of the liver with an intercostal approach at the 7th or 8th intercostal space. In B-mode an area was chosen without artefacts of ribs and without larger vessels. Two different methods of data acquisition were used. In the first method, we conducted ten separated acquisitions with the standard size SWE-box (22x18mm) as the pre-set and a single ROI measurement per scan. In the second approach the size of the SWE-box was increased to maximum (56x50mm) and ten ROI were selected in the SWE-box. The SWE-box was placed with the upper border 15–20 mm below the liver surface. In both approaches the real-time SWE modus ("Multi-Shot") was used and the image was paused after the whole SWE-box was colour filled and the propagation map showed parallel wave fronts. The measurements for each method were obtained with a circle shaped ROI of 10 mm diameter which was placed manually by the operator in an area of lowest standard deviation (SD) and parallel wave fronts in the propagation map. The ROI never overlapped each other. An example of the methods is given in Figs 1 and 2. The calculated value and the SD in m/s and kPa for each ROI were recorded and the mean and SD of ten measurements were calculated. The operator had more than three-years of experience in abdominal ultrasound exams and more than one year of experience in real-time elastography studies. The training and supervision of ultrasound and SWE was conducted by a highly experienced operator with more than forty years of experience in abdominal ultrasound exams and more than twelve years of experience in ultrasound based elastographic techniques.

### Statistical analysis

All statistical tests were done with SPSS Statistics version 27.0.0.0 (IBM, Armonk, USA). For comparison of the two methods a Bland-Altman plot was drawn and the interclass correlation coefficient (ICC) was calculated. The Bland-Altman plot shows the difference of two measurements, which is plotted against the mean of the two measurements [12]. In this plot the 95%-

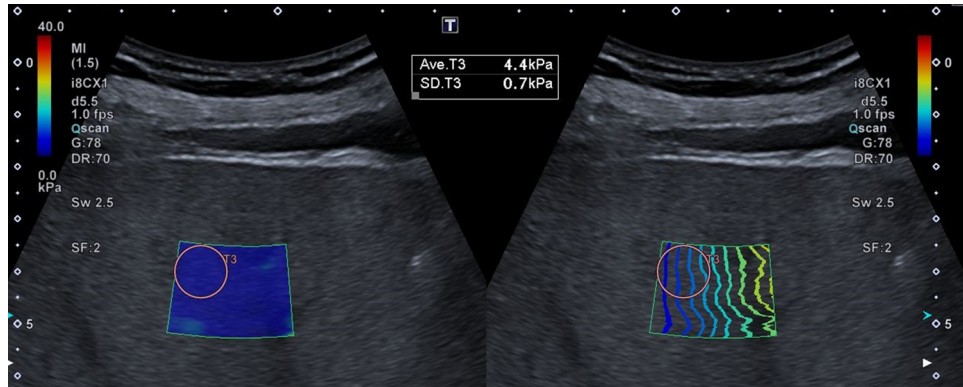

**Fig 1. Example of single ROI methods.** The example picture of one of the ten acquisitions with a single ROI for measurements. The shear wave elastography (SWE)-box is shown on the left side and the corresponding propagation map on the right. Figs 1 and 2 are examples from the same patient.

confidence interval is given as dotted lines. ICC estimates and their 95% confident intervals were calculated based on a mean-rating (k = 2), absolute-agreement, 2-way mixed-effects model. The inter-method agreement was classified as poor (ICC < 0.50), moderate (ICC = 0.50–0.75), good (ICC = 0.75–0.90) or excellent (ICC > 0.9) [13].

## Results

### Demographics

Sixty-nine volunteers were included in this study. For three participants (4%) the 2D-SWE failed because of obesity (BMI of more than 37 kg/m$^2$). The demographics of the 66 successful examined participants are as follows. Fifty-five women took part (83%). The mean age was 62 with a SD of ±13 years and a range of 22 to 86 years. The mean BMI was 25.3 with a SD of ±4.8 kg/m$^2$ and a range of 16 to 40 kg/m$^2$. The ultrasound was performed to monitor the primary

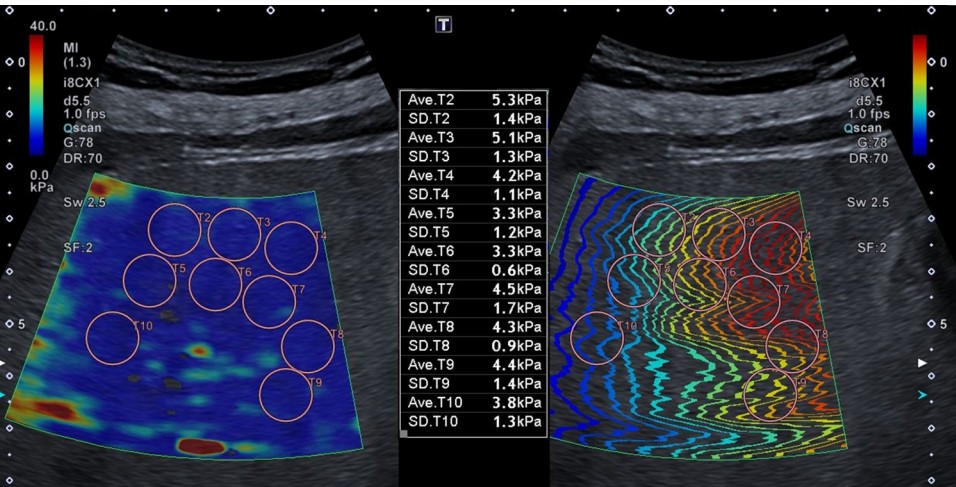

**Fig 2. Example of the ten ROI in one acquisition.** Example picture of one acquisition with ten measurements in the enlarged SWE-box. On the left side the SWE-box with colour-coded elastographic measurements and on the right side the corresponding propagation map are shown. Note that the Aplio ultrasound system does not show the first region of interest (ROI) after setting the tenth. Figs 1 and 2 show the results of an 86-year-old male patient with haemochromatosis, mean results are 1.16 m/s (3.9 kPa) and 1.21 m/s (4.3 kPa) for the ten acquisitions and one acquisition with ten measurements respectively.

**Table 1. Statistical data of both methods.**

|  | Ten single acquisitions, single ROI | One acquisition, ten ROIs |
| --- | --- | --- |
| **n** | 66 | 66 |
| **Mean** | 1.300 m/s (5.16 kPa) | 1.301 m/s (5.17 kPa) |
| **Mean standard deviation** | 0.156 m/s (1.39 kPa) | 0.133 m/s (1.28 kPa) |
| **Minimal value** | 1.083 m/s (3.37 kPa) | 1.115 m/s (3.53 kPa) |
| **Median value** | 1.265 m/s (4.84 kPa) | 1.280 m/s (4.94 kPa) |
| **Maximum value** | 1.749 m/s (9.33 kPa) | 1.807 m/s (11.24 kPa) |

Table 1 shows the statistical data compared for both methods. The difference in mean is not statistically significant (p = 0.9).

biliary cholangitis (33, 50%), primary sclerosing cholangitis (11, 17%), autoimmune-hepatitis (9, 14%), haemochromatosis (5, 8%), Wilson's disease (3, 5%), non-alcoholic-fatty-liver (3, 5%) or autoimmune cholangitis (2, 3%).

## SWE measurement

The statistical analysis of the mean value of each participant separated in both methods is shown in Table 1. The difference in mean is not statistically significant (p = 0.9). The method of one acquisition has a slight smaller range compared to ten single acquisitions with a lower maximum and a higher minimal value.

## Intermethod comparability

According to the Bland-Altman plot, the mean difference between the two methods was -0.0016 m/s with a SD of 0.121 and upper and lower limits of agreement of 0.206 and -0.209 respectively. The plot is shown in Fig 3. The ICC is 0.85 with lower and upper value of the 95[th]

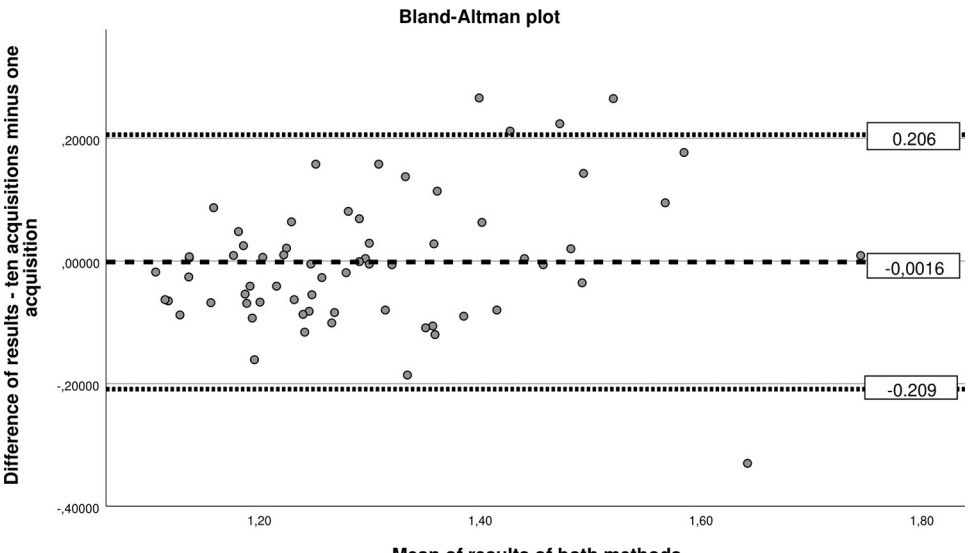

**Fig 3. Bland-Altman plot for comparison of individual measurements and the grouped-measurement method.** The horizontal axis shows the mean of both measurements and the vertical axis shows the difference between both measurements drawn. The red dotted lines are the upper and lower limits of agreement and the red stashed line is the mean of all differences. The plot shows the spread between the results of both methods which increases with higher mean values.

confidence interval of 0.75 and 0.91 respectively. According to Koo and Li it is a good reliability for an ICC [13].

## Discussion

Grouping ten ROIs in one acquisition allows a simpler examination in shorter time compared to multiple acquisitions and a single ROI and is comparable in mean and median value with a good ICC.

To simplify the procedure and reduce the amount of required time, one study showed that the mean or the median of five measurements are comparable to ten measurements [14]. Another study shows a good reliability for the median of just three valid measurements [15]. Both studies reduce the number of measurements, which increases the risk of statistical uncertainty and sampling bias. Current guidelines recommend three to five measurements and acquisitions respectively [2, 16, 17].

To the best of our knowledge exist only two studies of the liver which use multiple ROI in one acquisition. In the latest report Chung et al. (2019) [9] used grouped measurements of two to three ROIs. In the other study by O'Hara et al. (2017) [18], two ROIs in one acquisition are used. Both did not increase the SWE-box size, but they did increase the ROI area in a single measurement as suggested previously by Thiele et al. to improve the validity of measurements [19]. The easiest approach is to increase the size of the ROI, which was done by Wang et al. in a different 2D-SWE ultrasound system, but this approach increases the median and the range of the results [20] and keeps a small SWE-box. Similar approaches with different SWE systems have been studied in lymph nodes [21] and breast lesions [22]. In these entities the maximal stiffness distinguished the best between malignant and benign lesions. This approach needs further examination to validate the applicability to determine the grade of liver fibrosis.

Increasing the size of the SWE-box as suggested in our study has beneficial side effects. The examiner does get a better overview and can choose areas with good to excellent quality of elastographic measurements. Additionally, it gives a better impression of the elastographic pattern of the liver because of a greater area of view. Multiple ROIs assure to measure different areas in the liver. Single acquisitions und measurements bear the risk to repeat the procedure on the same area of liver, because of the limited anatomical options to perform a good SWE measurement. In this study the good correlation could be shown for values below 1.6 m/s equal to 7.7 kPa, because most of our participants are well treated outpatients with low grades of fibrosis ($\leq$F2). At higher values the spread between both methods increases, which is seen in most elastographic studies as well. Further research is needed to compare both methods for patients with higher grade of fibrosis or cirrhosis. Please note that the patients have mixed causes of liver disease, which could have influence on the kind of fibrotic changes and the results of the elastography.

## Conclusion

To conclude, maximizing the SWE-box and setting multiple ROIs is as good as performing ten acquisitions to confirm no or low grades of fibrosis (below 1.6m/s or 7.7 kPa, F0/1 fibrosis). In practical terms the suggested method shows more area of the liver tissue and the corresponding elastographic pattern and at the same time it is simpler and faster for daily routine. For higher grades of fibrosis ($\geq$F2) further studies are needed to be performed to prove the applicability of this method.

## Supporting information

**S1 Data.**
(XLSX)

## Author Contributions

**Conceptualization:** Moritz Weiher.

**Data curation:** Moritz Weiher.

**Formal analysis:** Moritz Weiher.

**Investigation:** Moritz Weiher, Felix Gunnar Richtering.

**Supervision:** Yvonne Dörffel, Hans-Peter Müller.

**Writing – original draft:** Moritz Weiher.

**Writing – review & editing:** Yvonne Dörffel, Hans-Peter Müller.

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
