## [Decision Letter · Decision Letter 0]

4 May 2022

PONE-D-21-33726Simplification of 2D shear wave elastography by an enlarged SWE box and multiple regions of interest in one acquisitionPLOS ONE

Dear Dr. Weiher,

Thank you for submitting your manuscript to PLOS ONE. After careful consideration, we feel that it has merit but does not fully meet PLOS ONE’s publication criteria as it currently stands. Therefore, we invite you to submit a revised version of the manuscript that addresses the points raised during the review process.

ACADEMIC EDITOR: Dear author, my feeling is that your work may be published in this journal after major revisions. In order to get this, please provide the following required changes:Editing and language revisions are needed.Revise and shorten Introduction and Discussion.==============================

We look forward to receiving your revised manuscript.

Kind regards,

Francesco Somma

Academic Editor

PLOS ONE

Additional Editor Comments:

The paper deals with the use of larger SWE-box and setting ten regions of interest (ROI) in one acquisition to improve the procedure and reduce the acquisition time.

The subject is timely and interesting. Reviewers remarked the need of editing and language revision. Moreover, reviewer 2 suggested to re-organize the section Discussion. I suggest to revise in a more focused way the section Introduction, too.

Reviewers' comments:

Reviewer's Responses to Questions

**Comments to the Author**

1. Is the manuscript technically sound, and do the data support the conclusions?

Reviewer #1: Yes

Reviewer #2: Partly

2. Has the statistical analysis been performed appropriately and rigorously? 

Reviewer #1: Yes

Reviewer #2: Yes

3. Have the authors made all data underlying the findings in their manuscript fully available?

Reviewer #1: Yes

Reviewer #2: Yes

4. Is the manuscript presented in an intelligible fashion and written in standard English?

Reviewer #1: Yes

Reviewer #2: Yes

5. Review Comments to the Author

Reviewer #1: Abstract

Line 26 : Please correct “Thru”

Introduction

Line 52-54 : Please rephrase

Line 55 : Please correct “collated”

Materials and methods

Line 64-65 : Please rephrase

Results

Linea 124 : Please correct “statistical”

Reviewer #2: Thank you for giving me the opportunity to review the article you submitted with the title "Simplification of 2D shear wave elastography by an enlarged SWE box and multiple regions of interest in one acquisition".

The topic of the manuscript refers to the use of larger SWE-box and setting ten regions of interest (ROI) in one acquisition to semplify the procedure and reduce the acquisition time.

The study presents data of primary scientific research and I found that the reported results have not been published elsewhere. The research meets all applicable standards for the ethics and research integrity. The article is presented in an intelligible fashion and adheres to appropriate reporting guidelines. Summary of results is given concisely are clearly supported by accurate and good quality images. Conclusions are appropriate and the results as well as the discussion supports them.

My comments on the manuscript are listed below:

- The discussion should be strengthened and in this regard we recommend multiple articles that have already dealt with this issue which in fact does not represent a particular clinical innovation.

Among the recommended articles:https://doi.org/10.1371/journal.pone.0265802.

- "Choosing ten different ROIs in one acquisition ALWAYS involves different parts of the liver" is not scientifically correct. Describe why you say this.

- The lateral ROI position contributed as much as the acquisition depth to the total variance in SWD. Locations close to the initial shear-wave excitation pulse were more robust to biases because of inaccurate probe – phantom coupling. The size of the ROI and acquisition box did not introduce significant variations.

How do you refute this statement?

- Many technical errors and need for vocabulary corrections are identified. A quick review is recommended.

In conclusion, despite my remarks and the inaccuracies found, the proposed study is substantiated and presented concisely. The stated aim is schieved. I would recommend the proposed article to be accepted for publication but after clearance of my comments. I would recommend the authors to pay more attention when preparing their manuscripts on methodology and punctuation.

6. PLOS authors have the option to publish the peer review history of their article (what does this mean?). If published, this will include your full peer review and any attached files.

Reviewer #1: No

Reviewer #2: No

---

## [Author Response · Author response to Decision Letter 0]

12 Jul 2022

ACADEMIC EDITOR: 

Dear author, my feeling is that your work may be published in this journal after major revisions. In order to get this, please provide the following required changes:

Editing and language revisions are needed.

Revise and shorten Introduction and Discussion.

Additional Editor Comments:

The paper deals with the use of larger SWE-box and setting ten regions of interest (ROI) in one acquisition to improve the procedure and reduce the acquisition time.

The subject is timely and interesting. Reviewers remarked the need of editing and language revision. Moreover, reviewer 2 suggested to re-organize the section Discussion. I suggest to revise in a more focused way the section Introduction, too.

----The introduction is shortened and the focus is set to a more technical point to address the reviewer’s comment that this method is not a clinical innovation.

Redundant sentences and phrases are deleted in the discussion. A new part with perspective in SWE in LN and breast lesions has been included, which opens an interesting point of view (maximal stiffness).

Reviewer #1: 

Abstract

Line 26 : Please correct “Thru” -- It is corrected.

Introduction

Line 52-54 : Please rephrase -- The lines are rephrased.

Line 55 : Please correct “collated” -- The correction is applied.

Materials and methods

Line 64-65 : Please rephrase -- The sentences are rephrased.

Results

Linea 124 : Please correct “statistical” -- It’s corrected.

 -- I apologize for the language mistakes and I am grateful for the kind remarks.

Reviewer #2: 

Thank you for giving me the opportunity to review the article you submitted with the title "Simplification of 2D shear wave elastography by an enlarged SWE box and multiple regions of interest in one acquisition".

The topic of the manuscript refers to the use of larger SWE-box and setting ten regions of interest (ROI) in one acquisition to simplify the procedure and reduce the acquisition time.

The study presents data of primary scientific research and I found that the reported results have not been published elsewhere. The research meets all applicable standards for the ethics and research integrity. The article is presented in an intelligible fashion and adheres to appropriate reporting guidelines. Summary of results is given concisely are clearly supported by accurate and good quality images. Conclusions are appropriate and the results as well as the discussion supports them.

My comments on the manuscript are listed below:

- The discussion should be strengthened and in this regard we recommend multiple articles that have already dealt with this issue which in fact does not represent a particular clinical innovation.

Among the recommended articles: https://doi.org/10.1371/journal.pone.0265802.

---- I revised the discussion. Redundancies were deleted and some aspects of the mentioned articles were included. The mentioned articles are quite interesting and expand my horizon, because I never dealt with lymph nodes nor mamma lesions. 

- "Choosing ten different ROIs in one acquisition ALWAYS involves different parts of the liver" is not scientifically correct. Describe why you say this.

I apologize for using incorrect wording and being redundant with the sentences before. The liver doesn’t get homogenous fibrotic. Like in histological examinations a sampling error exists. By choosing a larger SWE-box you gain a better overview and avoid to acquire elasticity in the same area of the liver by mistake.

- The lateral ROI position contributed as much as the acquisition depth to the total variance in SWD. Locations close to the initial shear-wave excitation pulse were more robust to biases because of inaccurate probe – phantom coupling. The size of the ROI and acquisition box did not introduce significant variations.

How do you refute this statement?

----- It's correct. Due to technical problems the signal from the shear wave gets more unreliable on its way through the liver. That depends partly because of hepatic fibrosis. In soft liver tissue the difference isn’t high as shown in this study. Therefore, it would be interesting if the results change in more fibrotic and cirrhotic livers.

- Many technical errors and need for vocabulary corrections are identified. A quick review is recommended.

---- I apologize for the language mistakes and corrections and reformulations have been applied.

In conclusion, despite my remarks and the inaccuracies found, the proposed study is substantiated and presented concisely. The stated aim is achieved. I would recommend the proposed article to be accepted for publication but after clearance of my comments. I would recommend the authors to pay more attention when preparing their manuscripts on methodology and punctuation.

---- Thank you very much for your kind remarks.

---

## [Editor Report · Decision Letter 1]

16 Aug 2022

Simplification of 2D shear wave elastography by an enlarged SWE box and multiple regions of interest in one acquisition

PONE-D-21-33726R1

Dear Dr. Weiher,

We’re pleased to inform you that your manuscript has been judged scientifically suitable for publication and will be formally accepted for publication once it meets all outstanding technical requirements.

Kind regards,

Francesco Somma

Academic Editor

PLOS ONE

---

## [Editor Report · Acceptance letter]

1 Sep 2022

PONE-D-21-33726R1 

Simplification of 2D shear wave elastography by enlarged SWE box and multiple regions of interest in one acquisition 

Dear Dr. Weiher:

I'm pleased to inform you that your manuscript has been deemed suitable for publication in PLOS ONE. Congratulations! Your manuscript is now with our production department. 

Kind regards, 

on behalf of

Dr. Francesco Somma 

Academic Editor

PLOS ONE